# Ovulation Induced by Intrauterine Seminal Plasma Increases Total Protein, PGE2, IL-8, and IL-1β in Uterine Fluid of Llamas (*Lama glama*)

**DOI:** 10.3390/ani13040554

**Published:** 2023-02-05

**Authors:** Alejandra Isabel Hidalgo, Cesar Ulloa-Leal, Gonzalo Gajardo, Gerardo López, Daniella Carretta, Rafael Agustín Burgos, Marcelo Ratto

**Affiliations:** 1Laboratory of Animal Reproduction, Faculty of Veterinary Sciences, Institute of Animal Science, Universidad Austral de Chile, Valdivia 5090000, Chile; 2Laboratory of Inflammation Pharmacology, Faculty of Veterinary Sciences, Institute of Pharmacology and Morphophysiology, Universidad Austral de Chile, Valdivia 5090000, Chile; 3Laboratory of Immunometabolism, Faculty of Veterinary Sciences, Institute of Pharmacology and Morphophysiology, Universidad Austral de Chile, Valdivia 5090000, Chile

**Keywords:** uterine immunity, inflammatory mediators, fertility, luteal phase, follicular phase, ovulation induction

## Abstract

**Simple Summary:**

Cytokines and prostaglandins are inflammatory mediators that are secreted by the immune cells but also by the female genital tract to regulate immune maternal tolerance. The balance of an adequate uterine immune response promotes reproductive success. No information is available regarding the characterization of inflammatory mediators in the uterine fluid of llamas during different stages of the reproductive cycle. Therefore, the main goal of the study was to determine the presence of these inflammatory mediators in the uterine fluid of llamas in the follicular and luteal phase induced using different stimuli of ovulation induction, among them, the intrauterine infusion of llama seminal plasma. Although there were inflammatory mediators present in both phases of the reproductive cycle, llamas in the luteal phase induced by seminal plasma presented the highest concentration of inflammatory mediators indicating that seminal plasma is rich in compounds that significantly influence the endometrial secretion of these inflammatory mediators.

**Abstract:**

The establishment of a state of immunotolerance in the female reproductive tract is important for embryo development, implantation and placentation. Llamas are induced ovulators and more than 98% of pregnancies occur in the left uterine horn. The objective of this study was to determine the uterine immune response of llamas in different stages of the reproductive cycle. Adult llamas (*n* = 20) were examined daily by transrectal ultrasonography to determine follicular growth and then randomly assigned to four groups: Follicular phase (*n =* 5); Luteal phase induced by an intramuscular administration of 50 ug of GnRH analogue (*n =* 5); Luteal phase induced by intrauterine infusion of seminal plasma (*n =* 5); and Luteal phase induced by mating (*n =* 5). Uterine fluid was collected separately from both uterine horns by non-surgical flushing to determine the presence of cells, total proteins and concentration of IL-1β, IL-6, IL-8, IFN γ, TNF-α and PGE2. Inflammatory cells were not observed in the uterine fluid and total protein pattern and inflammatory mediators did not differ between the left and the right horn amongst groups. Llamas treated with an intrauterine infusion of seminal plasma showed the highest concentration of total proteins, inflammatory cytokines PGE2, IL-8 and IL-1β in the uterine fluid. In conclusion, seminal plasma is made up of significant numbers of signaling molecules that are able to modify the uterine immune response in llamas.

## 1. Introduction

There is a close relationship between immunity and reproduction; the immune system may not only unravel uterine infections [1,2], but can also provide a state of immune tolerance required for implantation, pregnancy and calving [3,4,5,6]. The uterine immune response must be capable of discriminating antigens from infectious microorganisms, embryo or spermatozoa [7], in addition to regulating fertilization, implantation and pregnancy, through the hormonal influence of estrogen and progesterone [8]. When semen is deposited in the female genital tract during mating, the spermatozoa activate the complementary cascade, generating neutrophil chemotaxis [9] together with an increase in the expression of inflammatory genes [10] that participate in the uterine inflammatory response. Uterine fluid is secreted from the endometrial and luminal glands and selectively transported from the blood through uterine epithelial cells [11]. It contains a complex mixture of growth factors, hormones, enzymes, carrier proteins, ions, lipids, glucose and amino acids that are involved in signal transduction, glycolysis, angiogenesis, protection against oxidative stress and immunity [12,13,14]. The uterine mucosa undergoes secretory changes during different physiological stages of the reproductive cycle [15]; therefore, the characterization of the composition of the uterine fluid would allow not only a better understanding of changes in the uterine environment during the different phases of the reproductive cycle but it could also be useful to identify markers of the reproductive health status of the individual [16].

Some inflammatory mediators are involved in the functionality of the reproductive system of the female and contribute to the success of fertility [17], amongst them PGE, IL-8, IL-6, IL-1β and TNF-α that are also synthesized by endometrial cells [18,19,20,21,22,23]; meanwhile, others, such as IFNγ, are only synthesized by immune cells [24]. These cytokines are involved, to some extent, in ovulation, embryo implantation and the early stages of embryonic development [25,26,27,28,29,30,31,32,33,34,35,36,37,38,39,40,41,42,43,44,45,46,47]. On the other hand, llamas are an induced ovulatory species with a distinctive reproductive physiology characteristic. For instance, they have a long gestation period (342–350 days) and more than 98% of pregnancies take place in the left uterine horn [48,49,50], regardless of the ovulation site [51]. In fact, embryo migration from the right to the left uterine horn must occur at day 8 post fertilization to avoid luteolysis and ensure the success of the pregnancy [52]. Therefore, it would be interesting to determine if changes in the concentration of inflammatory mediators in the uterine fluid could have any relevance to the process of embryo migration, which arises on day 8 after ovulation. 

The aim of the study was to determine the presence of immune cells, total proteins and the concentration of inflammatory mediators in the uterine fluid of llamas during the follicular and luteal phases induced by GnRH analogue, intrauterine infusion of seminal plasma and natural mating. In addition, the presence and concentration of these inflammatory mediators were compared between the left and right uterine horn.

## 2. Materials and Methods

### 2.1. Animals and Housing

The study was conducted in the llama farm of the Department of Animal Science at the Universidad Austral de Chile, Valdivia, 39°48′13.6″ S–73°15′08.6″ W and 15 m above sea level. Llamas have free access to pasture throughout the day and they were supplemented twice a day with hay and pellet (dry matter 18% minimum, crude protein 16% minimum, crude fiber 14% maximum, ethereal extract 3% minimum Champion^®^ (Champion S. A., Peñaflor, Chile), and water ad libitum. All experiments were conducted in accordance with the Guidelines for the use of animals in experimentation of the Universidad Austral de Chile, and the National Guidelines on the Use of Experimental Animals of the “Comisión Nacional de Ciencia y Tecnología de Chile” law No 20.380. All animal experiments were also approved the Institutional Ethic Review Committee (No. Bioethics Report 339/2019 and No. biosafety report 0004-19).

### 2.2. Experimental Design

Non-pregnant and non-lactating adult llamas (*n =* 20), between 6 and 8 years old and weighing 130 to 150 kg were used in this study. These llamas were examined daily by transrectal ultrasonography using a 7.5 MHz transducer coupled with a MyLab 30 scanner (MyLab 30; Esaote, Maastricht, The Netherlands) to determine follicular growth. Females exhibiting a follicle ≥ 8 mm in diameter that grew for 3 consecutive days were randomly assigned to the following groups:iFollicular phase: (*n =* 5) based on previous studies [51,53] the presence of a follicle ≥ 8 mm in diameter for 3 consecutive days is an indicator of the follicular phase and it is associated with sexual receptivity. Ovulation will not occur unless females are given ovulatory stimuli (hormones or mating). The females of this group were submitted to uterine flushing at the third day of scanning corroborating the presence of the dominant follicle;iiLuteal phase induced by GnRH: (*n =* 5) females were given an i.m. dose of 50 ug analogue of GnRH gonadorelin acetate Gonasyl^®^ (Syva, León, Spain);iiiLuteal phase induced by intrauterine infusion of seminal plasma: (*n =* 5) females were intrauterine-infused with 5 mL of llama seminal plasma with an insemination catheter accoupled to a 20 mL syringe using the transrectal palpation technique;ivLuteal phase induced by mating: (*n =* 4) females were mated with fertile males, successful copulation was confirmed by the introduction of the penis into the female genital tract, with a minimum duration of the copulation of 25 min.

Llamas from the induced luteal phase groups were examined by ultrasonography every 12 h to confirm ovulation. Ovulation was defined as the disappearance of the dominant follicle and the presence of a corpus luteum at Day 8 (Day 0 = ovulation). These females were also submitted for uterine flushing at Day 8 after ovulation. 

### 2.3. Llama Male Semen Collection for Preparation of Seminal Plasma

Semen was collected twice per week using an artificial sheep vagina from 4 mature male llamas kept at the Institute of Ciencia Animal, UACH (39°38′ S–73°5′ W and 19 m above sea level) as previously reported [54]. Semen was centrifuged for 30 min at 2500× *g* at room temperature and the supernatant, seminal plasma, was aspirated with a plastic sterile plastic pipette, transferred to a 15 mL Falcon tube and finally stored at −20 °C until use. An average of 15 ejaculates were collected from each male. A cell smear of the supernatant was made to confirm the absence of cells and spermatozoa. 

### 2.4. Uterine Fluid Collection by Non-Surgical Uterine Flushing 

Llamas were given an epidural anesthesia with a total dose of 60 mg of lidocaine chlor-hydrate contained in 3 mL volume (Lidocalm; Drag Pharma, Quilicura, Chile). A Foley catheter, 16 Fr (Medicoplast, Illingen, Germany), was guided by rectal palpation through the cervix to the uterine horn. The cuff of the catheter was inflated with 4–5 mL of air and fixed in the third cranial part of the uterine horn, adding 1 or 2 mL more if required. Each uterine horn was flushed separately using a different sterile catheter with a 5 mL volume of phosphate buffer saline solution accoupled to a 15 mL syringe. The fluid was collected by aspiration (negative pressure) with the same syringe.

### 2.5. Cytology Analyses of Uterine Fluid

A 50 µL aliquot of freshly collected uterine fluid from both horns was placed on a slide and allowed to dry at room temperature and smears were performed for cellular characterization. Subsequently, cell staining was performed with the Hemacolor^®^ microscopy kit (Merck, Darmstadt, Germany) and the samples were analyzed using an Olympus BX51 microscope (Olympus, Tokyo, Japan).

### 2.6. Protein Quantification and Characterization Pattern of Uterine Fluid

The freshly extracted uterine-fluid samples from both horns were centrifuged at 1000× *g* for 15 min at 4 °C and the supernatant was stored at −80 °C until processing. Protein quantification was performed with the Qubit^®^ Protein Assay Kits (Thermo Fisher Scientific, Waltham, MA, USA) and 30 µg of the total proteins were separated by electrophoresis on 12% (SDS-PAGE) gel at 100 V for 2 h. Gel staining was completed with 10% Coomassie blue. When the adequate degree of staining was reached, the gels were washed with 0.1% TBS-Tween^®^ 20 buffer solution (10 mM Tris-HCl, 68 mM NaCl, and 0.1% Tween^®^ 20). The analysis of the gels was performed with Image 1.51j8 software.

### 2.7. Measurements of IL-1β, IL-6, IL-8, IFN γ, TNF-α and PGE2 Concentration in the Uterine Fluid

Aliquots of uterine fluid from both uterine horns were thawed and used to estimate the concentration of all inflammatory mediators. IL-1b and IL-6 were measured using a bovine IL-1β ELISA Kit (#ESS0027, Thermo Fisher Scientific, MA, USA) and IL-6 (#ESS0029, Thermo Fisher Scientific, MA, USA), according to the manufacturer’s instructions. Briefly, the capture antibody was incubated overnight; wells were then blocked for 1 h; subsequently, 200 μL of sample was added and incubated for 1 h. After the plates had been washed twice, the detection antibody was added and incubated for 1 h. After a further two washes, streptavidin was added, and the mixture was incubated for a further 30 min. Following this, wells were incubated for 20 min in the dark by substrate solution and stop solution was added. All procedures were performed at room temperature. Finally, the samples were analyzed at 450 nm and 550 nm in an automatic Varioskan Flash Reader (Thermo Fisher Scientific, MA, USA). 

For the IL-8 analysis, an ELISA Kit (CXCL8) (#3114-1A-6; MABTECH, Stockholm, Sweden) was used, according to the manufacturer’s instructions. Briefly, in a 96 well plate, 100 μL of capture antibody was incubated overnight at 4 °C. Then, 2 washes were performed and blocked for 1 h. Five washes were carried out again, the samples were loaded and incubated for 2 h. Biotinylated monoclonal antibody were incubated for 1 h. Five washes were carried out, and streptavidin-din-ALP were added and incubated for 1 h. To end, five washes were performed and p-nitrophenyl-phosphate (p-NPP) was added to analyze in at 405 nm in an automatic Varioskan Flash Reader (Thermo Fisher Scientific, MA, USA).

For the IFNγ analysis, a DuoSet^®^ ELISA (#DY2300; R&D Systems. Inc, Minneapolis, MN, USA) was used and for the TNF α analysis a DuoSet^®^ ELISA (#DY2279; R&D Systems. Inc, USA and Canada) was used, both according to the manufacturer’s instructions. Briefly, 100 μL capture antibody was incubated overnight, then wells were washed and blocked by 300 μL reagent diluent at 1 h. Washing was repeated, and samples were incubated at 2 h. Washes were repeated, and detection antibodies were incubated at 2 h. Again, washes were performed, and streptavidin HRP was incubated at 20 min. After a new wash substrate solution was incubated for 20 min stop solution was added. Immediately, the optical density was determined using an automatic Varioskan Flash Reader (Thermo Fisher Scientific, MA, USA) set to 450 nm, corrected and set to 540 nm.

For the PGE2 analysis, a Prostaglandin E2 Assay (# KGE004B; R&D Systems. Inc, USA and Canada) was used according to the manufacturer’s instructions. Briefly, in a 96 well plate, 150 μL of sample and 50 μL of Primary Antibody Solution were combined and incubated for 1 h at room temperature on a horizontal orbital microplate shaker. Then, 50 μL of PGE2 Conjugate was added and incubated for 2 h at room temperature with agitation. The plate was washed and substrate solution was added and incubated for 30 min. Finally, the reaction was finished with stop solution, followed by an analysis set to 450 nm, corrected and set to 540 nm in a Varioskan Flash Reader (Thermo Fisher Scientific MA, USA).

### 2.8. Statistical Analysis 

The concentration of total protein and inflammatory mediators was compared between the treatment groups and between the left and the right uterine horns of each treatment group, using a non-parametric *t*-test and One-Way ANOVA because the data did not show a normal distribution and homogeneity of variance according to the tests of Shapiro–Wilks and Bartlett, respectively. A value of *p* < 0.05 was considered significant. Data were represented on a bar graph as mean ± S.E.M. All analyses were performed using GraphPad Prism^®^ software (v 5.0; GraphPad Software, CA, USA).

## 3. Results

There was no presence of inflammatory cells in the uterine fluids collected from females during the follicular and luteal phases, only a slight presence of epithelial cells on the uterine mucosa (Figure 1). 

The protein quantification and characterization pattern of uterine fluid by group and from each the right and the left uterine horn, collected during the follicular and luteal phases are shown in Figure 2. An average of 16 Coomassie blue-stained bands were observed in the polyacrylamide gel of the uterine fluid of the llamas (Figure 2A). There were no differences between the number of bands observed in the uterine fluid between the left and the right horn (Figure 2B). In all the llamas evaluated, the presence of bands of 100, 70, 50 and 40 KDa were detected. However, the protein concentration was significantly higher in uterine fluid collected from the luteal phase induced by seminal plasma than that of the follicular phase (Figure 2C). When total protein concentration was individually analyzed between the left and the right uterine horn, it was higher in the uterine fluid of the right horn from the seminal plasma-treated females than that of the right horn from those in the follicular phase (Figure 2D). 

Inflammatory mediators were identified in the uterine fluid of llamas from the follicular and luteal phases, both as treated groups and as the left and the right uterine horns of each treated group. Significant changes in the concentration of PGE2, IL-8 and IL-1β were observed, while TNF-α, IFNγ and IL-6 were similar between groups (Figure 3). The concentrations of PGE2, IL-8 and IL-1β were higher during the luteal phase in those females treated with an intrauterine infusion of seminal plasma. It should be noted that the concentration of PGE2 was significantly higher in the females of the luteal phase induced with seminal plasma than those llamas in the follicular phase and the luteal phase induced with GnRH and mating (Figure 3A). The IL-8 concentration of llamas in the luteal phase induced with seminal plasma was significantly higher than llamas induced by GnRH and mating (Figure 3B). There was only a significant increase in IL-1β in luteal phase llamas induced with seminal plasma relative to those induced by GnRH (Figure 3C).

Concentration of PGE2, IL-8, IL-1β, IFNγ and IL-6 in the uterine fluid between left and right horns did differ among experimental groups but it did not change within the same group (Figure 4A–E).

## 4. Discussion

This study describes for the first time the concentration of inflammatory mediators present in the uterine fluid of llamas during the follicular and the luteal phases. Immune cells were not found in the uterine fluid of all groups (Figure 1), as previously described in humans [1,55] and in other animals [56,57] because immune cells are located in the uterine endometrium and myometrium, where they secrete their products into the uterine lumen. Meanwhile, unlike abundant immune cells, the endometrial epithelium cells are commonly found in the uterine fluid when the endometrium is exposed to pathogens or semen or after childbirth [58,59,60,61]. In the present study, immune cells were not observed in any of the groups, including those females treated with intrauterine infusion of seminal plasma or natural mating since sampling was performed 8 days after treatment so there is no longer any immune cells present in the uterine lumen [9]. 

The protein pattern observed in the uterine fluid of llamas (Figure 2A) presented a lower number of bands similar to that previously reported for alpaca uterine fluid [12] where 21 and 25 protein bands were observed from the right and the left horn of pregnant and non-pregnant females, respectively. The total protein concentration of the uterine fluid from the right and left horn of the llamas was higher than that previously described in alpacas [12], perhaps this higher concentration may be related to the methodology of sampling, as in the present study the uterine fluid was collected from in vivo instead of slaughterhouse material [12]. 

The inflammatory mediators analyzed in this study are related to the process of ovulation, luteinization and implantation in mammals; however, it is the first time that they have been described in the uterine fluid of llamas. Females treated with an intrauterine infusion of seminal plasma presented a high concentration of PGE2, IL-8 and IL-1β, which reinforces the modulatory role of seminal plasma to establish an immune tolerance process in the female to favor embryonic development and implantation [9], the concentration of these inflammatory mediators has also been found to increase in bovine endometrial epithelial cells stimulated with seminal plasma [62,63]. In addition, studies conducted in mice have shown that seminal plasma enhances implantation through mechanisms associated with PGE2 [64] and increases the production of IL-8 in bovine [65] and ovine [66] endometrial epithelial cells. In addition, it has been reported that seminal plasma improves IL-1β expression in mouse and human endometrium [67,68]. However, the concentration of these inflammatory mediators did not increase in llamas after natural mating. Although the reason for this phenomenon is not well understood, a plausible explanation can rely on results reported on in vitro studies [69], where spermatozoa are able to reduce the expression of IL-6, IL-8 and TNF-α in porcine uterine epithelial cells. It would therefore be interesting to evaluate the effect of only intrauterine sperm infusion on the production of inflammatory mediators in uterine fluid and ovulation induction in this species. The inflammatory mediators detected in the uterine fluid were produced by the endometrium and do not correspond to those that are part of the seminal plasma considering that uterine samples were collected 8 days after ovulation and that these inflammatory mediators have a short half-life [70,71,72,73].

The higher PGE2 concentration observed in llamas aligns with previous results found in bovines [74] and gilts [75] during the periovulatory and implantation periods, respectively, where an increase in the production and expression of PGE2 in the uterine fluid was found. Similarly, there was a significant increase in the pro-inflammatory mediators IL-8 and IL-1β that could also be associated with endometrial receptivity during embryo implantation [36,76]. In addition, the presence of IL-8 could induce a uterine immune response, through a possible participation of neutrophils, similar to what occurs in the porcine endometrium, where neutrophils increase migration under high expression of IL-8 after insemination [69]. No information regarding inflammatory mediators related to the uterine immune system has been reported in other induced ovulatory species such as camelids, except for the IL-8 and IL1β characterization described in rabbits [34,77]. The concentrations of INFγ, TNFα and IL-6 in the uterine fluid of llamas did not change among groups, a feature that may be due to the fact that these cytokines are mainly associated with chronic uterine infections and inflammatory processes. Although INFγ is not produced by the endometrium, it may be related to endometritis in mares [78] and preterm birth in women [79]. TNFα concentration increases in the uterine fluid of women during endometrial receptivity [76] and has also been detected in the uterine fluid of mice [80]. IL-6 concentration is high in the uterine fluid of sows during embryo implantation [75], in preterm parturition in women [80], and in mares with endometritis [78] and it is a marker of endometriosis in the uterine fluid of women [81]. 

The increase in cytokines in the uterine fluid during the luteal phase could be influenced by the effect of progesterone on endometrial tissues, since during the luteal phase there is a greater number of progesterone receptors on endometrial perivascular cells [82], that regulate the expression of genes related to the inflammatory response [83] and the expression of cytokines [84]. It was observed that the luteal phase of llamas given GnRH analogue did not increase cytokines in the uterine fluid. Although GnRH has a modulatory effect on cell proliferation in endometrial cancers in humans [85], its pro-inflammatory effect on uterine tissue is unknown. GnRH decreases the production of cytokines, through a direct effect on the endometrium; additionally, it has been described [86,87] as the expression of GnRH receptors in the endometrium, hence the in vitro culture of endometrial cells treated with GnRH diminished the expression of IL-8. In addition, GnRH-treated llamas lack the interaction with seminal plasma components that could also be responsible for the increased concentration of uterine cytokines, which could also suggest that post-ovulation cytokine production occurs primarily through the uterine level.

## 5. Conclusions

The uterine immune response of llamas by the presence of inflammatory mediators in the uterine fluid confirms the relationship between reproduction and immunity, to ultimately promote embryo development and implantation. Based on the results of the present study, it can be concluded that there is a differential profile of inflammatory mediators in the llama uterine fluid during the follicular and luteal phases; however, there was not a differential production of these mediators between the right or the left uterine horns. Intrauterine infusion of llama seminal plasma induces ovulation and generates a higher concentration of PGE2, IL-8 and IL-1β in the uterine fluid during the luteal phase. Therefore, it would be important to consider the impact of different ovulation-inducing treatment on uterine response and the outcome of successful pregnancy in llamas. 

## Figures and Tables

**Figure 1 animals-13-00554-f001:**
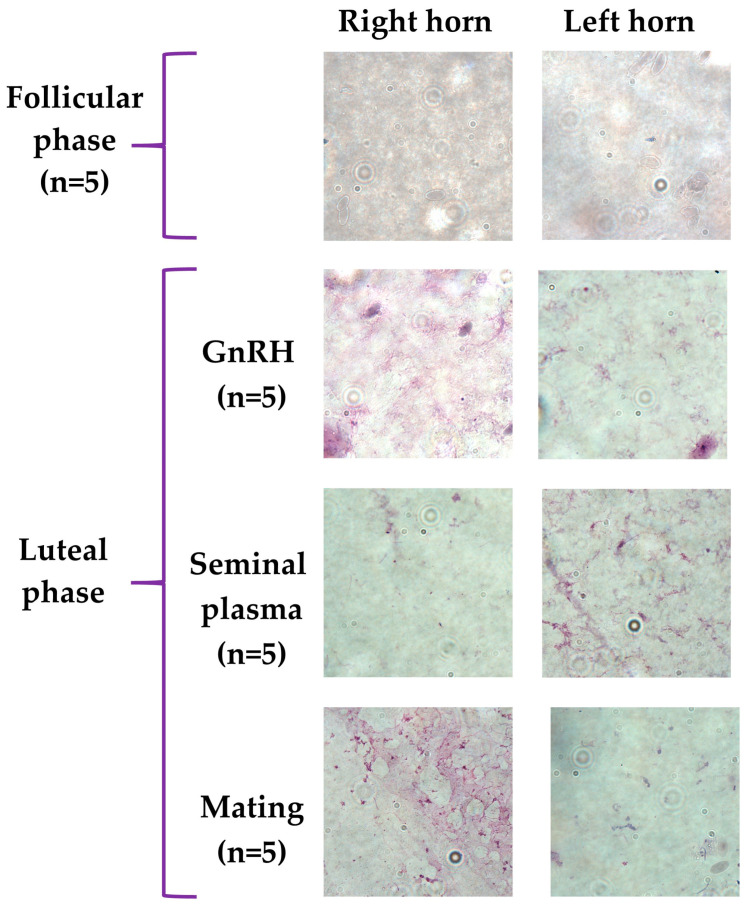
Absence of inflammatory cells in uterine fluid of llamas in follicular and luteal phase. A 50 μL aliquot of fresh uterine fluid from uterine horns. Right and left smears were performed for cellular characterization. Hemacolor^®^ microscopy kit (Merck, Germany) was used for cell staining. The images were produced with the Olympus BX51 microscope (Olympus, Japan). The smears were observed in magnification 100X and with 10 fields per sample.

**Figure 2 animals-13-00554-f002:**
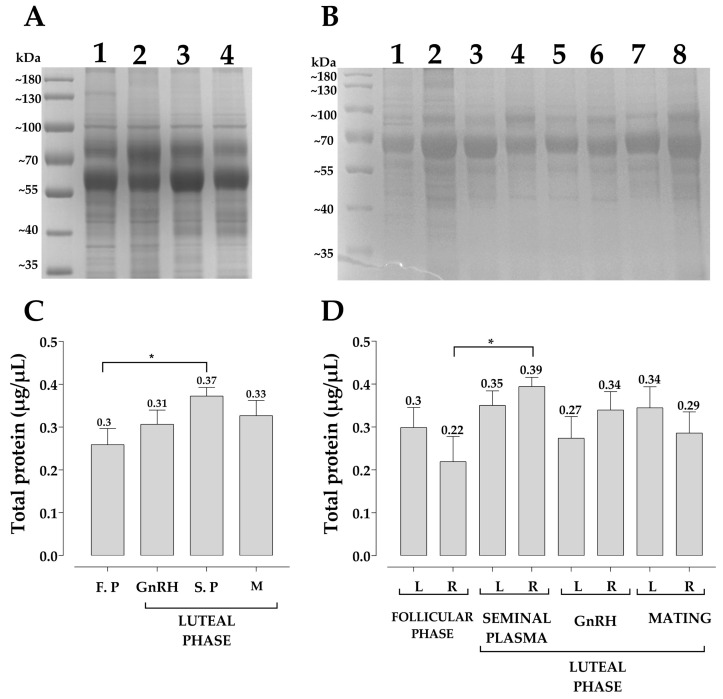
Total protein concentration in fluid uterine is higher in llamas induced with seminal plasma. The quantification of total proteins was carried out by fluorescence measurement. A total of 30 µg of total proteins were separated by electrophoresis on 12% SDS-PAGE gel. Gel staining was completed with 10% Coomassie blue (**A**,**B**). Lanes 1: follicular phase; 2: GnRH-induced luteal phase; 3: Seminal Plasma-induced luteal phase; 4: Mating-induced luteal (**A**). Lanes 1: left horn of follicular phase; 2: right horn of follicular phase; 3: left horn of luteal phase by seminal plasm; 4: right horn of luteal phase by seminal plasm; 5: left horn of luteal phase by GnRH; 6: right horn of luteal phase by GnRH; 7: left horn of luteal phase by mating; 8: right horn of luteal phase by mating (**B**). Representative bar graph of total protein concentration of uterine fluid, llamas in follicular phase and subjected to ovulation induction with GnRH, seminal plasma and mating (**C**,**D**). Analysis of uterine fluid of each uterine horn (**C**). Analysis of uterine fluid of each study group (**D**). Each bar represents the mean ± SEM of at least five independent experiments. F. P = follicular phase; S. P = seminal plasma; M = mating; L = left; R = right. * *p* < 0.05.

**Figure 3 animals-13-00554-f003:**
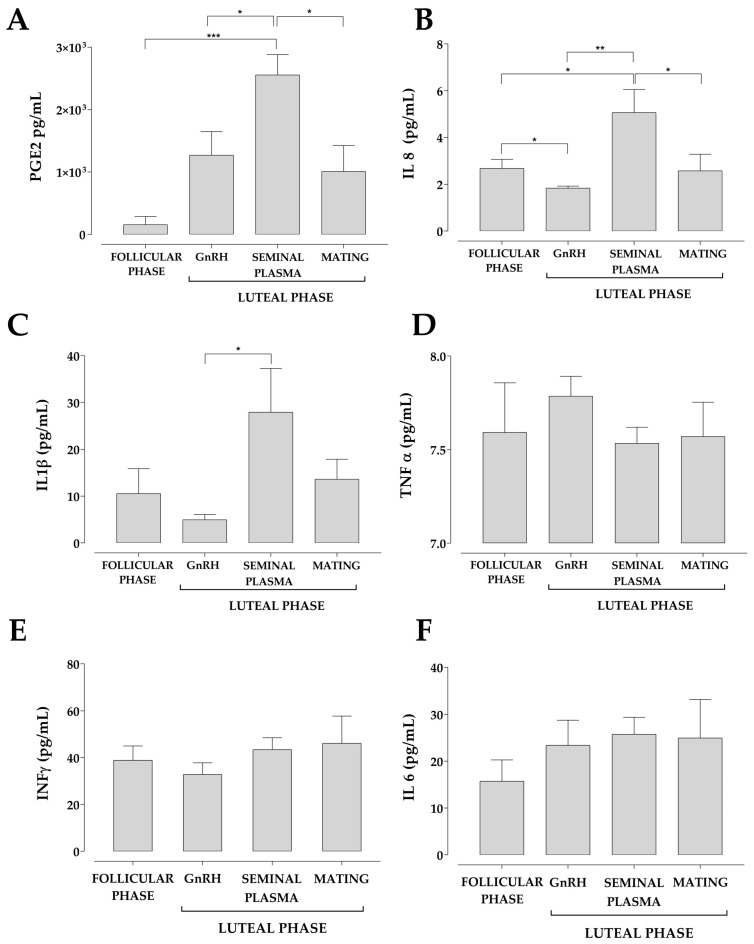
Induction of ovulation with seminal plasma induced an increase in PGE2, IL-8 and IL-1β concentration in the uterine fluid of llamas. Representative bar graph of the concentration of PGE2 (**A**), IL-8 (**B**), IL-1β (**C**), TNF-α (**D**), INF γ(**E**) and IL-6 (**F**). Concentration of inflammatory markers were measured using ELISA test in triplicate. Each bar represents the mean ± SEM of at least five independent experiments. * *p* < 0.05; ** *p* < 0.01; *** *p* < 0.001.

**Figure 4 animals-13-00554-f004:**
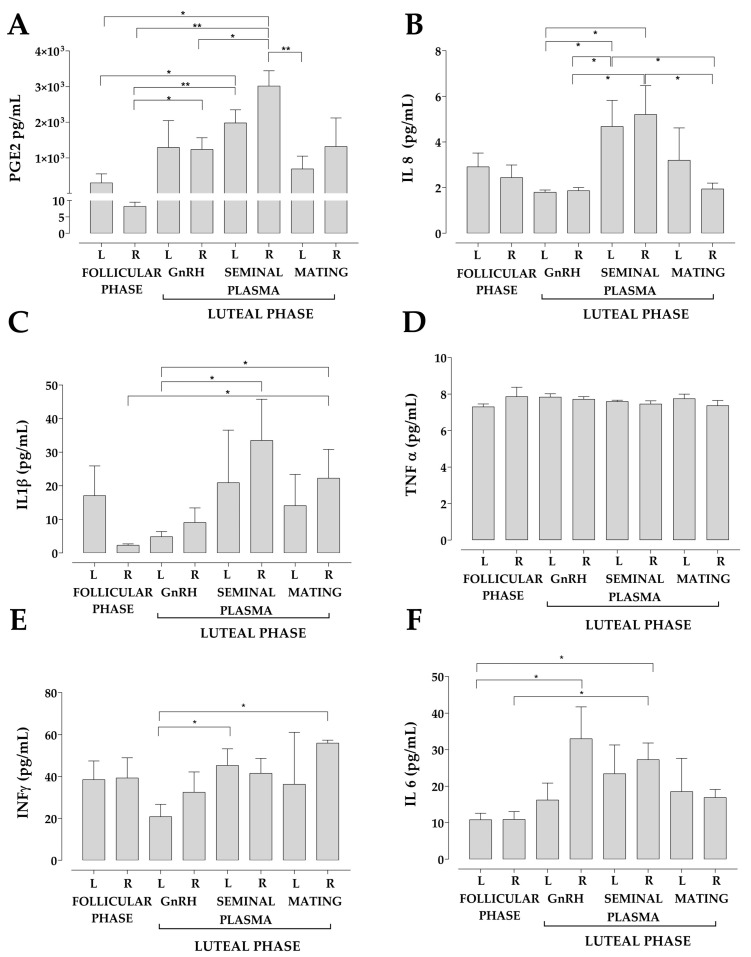
The concentration of PGE2, IL-8, IL-1β, INF γ and IL-6 varied significantly between the left and the right uterine horn of some study groups. Representative bar graph of the concentration of PGE2 (**A**), IL-8 (**B**), IL-1β (**C**), TNF-α (**D**), INF γ(**E**) and IL-6 (**F**). Concentration of inflammatory markers was measured using ELISA test in triplicate of at least five independent experiments. Each bar represents the mean ± SEM. L = left; R = right. * *p* < 0.05; ** *p* < 0.01.

## Data Availability

The data presented in this study are available on request from the corresponding author. The data are not publicly available to preserve privacy of the data.

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
