# Peer review of "Ovulation Induced by Intrauterine Seminal Plasma Increases Total Protein, PGE2, IL-8, and IL-1β in Uterine Fluid of Llamas (Lama glama)"

_animals, 2023, doi:10.3390/ani13040554_

Round 1

Reviewer 1 Report

Minor comments:

1.     Page 2, line 54: reference [3]-[6], should be [3-6].

2.     Page 2, line 57: replace “se-men” with “semen”.

3.     Page 6, line 221: replace “50 µl” with “Fifty microliters”.

 Major comments:

1.     One of the major shortcomings of the manuscript is that a technical method/s to prepare/produce seminal fluid was not presented in the Method. As a result, it is unclear if the seminal fluid was collected from one ram or it was pooled from several ram, and fresh or frozen.

2.     The experimental design had some flaw, such as no control from sperm, therefore the authors cannot be explained the reason for the results of inflammatory mediators did not increase in llamas after natural mating.

Reference: Spermatozoa and seminal plasma induce a greater inflammatory response in the ovine uterus at oestrus than dioestrus.

Scott JL, Ketheesan N, Summers PM.

Reprod Fertil Dev. 2009;21(7):817-26. doi: 10.1071/RD09012.

Author Response

Dear reviewer 1.

PDF file with point-by-point response is attached.

Best regards

Reviewer 2 Report

Results in this paper indicated that seminal plasma is rich of compounds that significant influence regulate the endometrial secretion of inflammatory mediators such as PGE2, IL-8, and IL-1β. It is well known that there are lots of molecules in seminal plasma, including RNAs, proteins, cytokines……maybe include PGE2, IL-8, and IL-1β in seminal plasma. Thus, the authors should demonstrate that inflammatory mediators (PGE2, IL-8, and IL-1β) increased in uterine fluid are derived from endometrium, not from seminal plasma.

 Pictures’ resolution needs to be improved and Figure 3C should be correct.

Author Response

Dear reviewer 2.

PDF file with the point-by-point is attached .

Best regards.

Round 2

Reviewer 1 Report

tidy up the reference format in the text, such as [1]-[2], should be [1, 2] etc.

Reviewer 2 Report

Accept in present form